# CSA-AKI: Incidence, Epidemiology, Clinical Outcomes, and Economic Impact

**DOI:** 10.3390/jcm10245746

**Published:** 2021-12-08

**Authors:** Alan Schurle, Jay L. Koyner

**Affiliations:** 1Department of Anesthesia and Critical Care Medicine, University of Chicago, 5841 South Maryland Ave., MC4028, Chicago, IL 60637, USA; 2Section of Nephrology, Department of Medicine, University of Chicago, 5841 South Maryland Ave., Suite S-507, MC5100, Chicago, IL 60637, USA; jkoyner@uchicago.edu

**Keywords:** acute kidney injury, cardiac surgery, epidemiology, outcomes, renal replacement therapy, mortality, clinical nephrology

## Abstract

Cardiac surgery-associated acute kidney injury (CSA-AKI) is a common complication following cardiac surgery and reflects a complex biological combination of patient pathology, perioperative stress, and medical management. Current diagnostic criteria, though increasingly standardized, are predicated on loss of renal function (as measured by functional biomarkers of the kidney). The addition of new diagnostic injury biomarkers to clinical practice has shown promise in identifying patients at risk of renal injury earlier in their course. The accurate and timely identification of a high-risk population may allow for bundled interventions to prevent the development of CSA-AKI, but further validation of these interventions is necessary. Once the diagnosis of CSA-AKI is established, evidence-based treatment is limited to supportive care. The cost of CSA-AKI is difficult to accurately estimate, given the diverse ways in which it impacts patient outcomes, from ICU length of stay to post-hospital rehabilitation to progression to CKD and ESRD. However, with the global rise in cardiac surgery volume, these costs are large and growing.

## 1. Introduction

Acute kidney injury (AKI) is currently defined by increases in serum creatinine and decreases in urine output over time. The Kidney Disease Improving Global Outcomes (KDIGO) consensus definition of AKI defines stage 1 AKI as a rise of ≥0.3 mg/dL within 48 h or an increase of ≥1.5 times baseline over 7 days, or urine output of <0.5 mL/kg for 6 h with the subsequent stages representing more severe kidney injury ([1], Table 1).

AKI occurs in approximately 20–30% of patients following cardiac surgery, and while there is no specific definition of cardiac surgery-associated AKI (CSA-AKI), clinicians apply the aforementioned KDIGO criteria. This high incidence of CSA-AKI reflects an interaction between patient co-morbidities and their peri- and intra-operative care [2,3,4]. While the development of AKI in this population typically occurs in older, sicker patients who require more complex surgeries, there are patients without these risk factors who still develop CSA-AKI [5]. Many of these established preoperative risk factors lead to longer operations with prolonged exposure to cardio-pulmonary bypass (CPB) and aortic cross clamping; an extended duration of inadequate circulation; and accompanying supportive care measures including inotropes, vasopressors, fluid and blood administration, and mechanical circulatory support [3].

Standardizing the definition of AKI into the KDIGO definition (and its predecessors) has advanced our understanding of AKI and improved epidemiologic AKI research. Newer biomarkers may be able to quickly and accurately diagnose kidney injury and specify patient phenotypes, including those whose AKI will persist (longer duration) or progress (worsen severity) [6,7,8,9]. There are growing calls for these biomarkers to be incorporated into routine clinical practice for use in risk stratification and diagnosis of AKI [10,11]. The importance of a deeper understanding of CSA-AKI is further magnified when considering the large economic impact and adverse patient outcomes associated with AKI [12,13]. In this review, we will examine the incidence and epidemiology of CSA-AKI, risk stratification including the importance of emerging biomarkers, various treatments and bundles, patient outcomes, and economic impact.

### 1.1. Risk Factors and Scoring

Risk factors of CSA-AKI can be categorized into patient and intra- and postoperative factors. Currently, a patient’s preoperative risk factors are largely non-modifiable and include demographics, such as age and gender; conditions such as hypertension, hyperlipidemia, diabetes mellitus, and vascular disease; and end-organ sequelae such as chronic kidney or liver disease, anemia, and previous stroke [3] (Table 2).

Many if not all of these risk factors also underlie the indications for cardiac surgery with chronic kidney disease, often being amongst the most prominent factors associated with CSA-AKI [14]. The patient’s preoperative hemodynamics and urgency of the surgical indication (e.g., emergency aortic dissection repairs or coronary artery bypass grafting) also increase risk, especially in conditions such as acute coronary syndrome and cardiogenic shock that require an intra-aortic balloon pump (IABP). The three commonly used scoring systems that consolidate these risk factors into predictive models for postoperative AKI and receipt of renal replacement therapy (RRT) are the Cleveland Clinic model, the Mehta Score, and the Simplified Renal Index [15,16,17]. These three systems use differing patient variables and have varying discrimination, but the Cleveland Clinic model has the highest level of discrimination in one validation study with an area under the ROC of 0.86 for postoperative RRT and 0.81 for postoperative stage 2 AKI by AKI-Network (AKIN) criteria (which is identical to KDIGO stage 2) [18] (Table 3).

While there is a focus on preventing severe and dialysis-requiring AKI because it is a major cost-burden and carries the highest morbidity and mortality, it is important to remember that over 90% of CSA-AKI is stage 1 (though this is also associated with adverse patient outcomes compared to those who do not develop AKI) [19].

Non-patient factors that confer greater risk of AKI include more complex operations, such as combination CABG and valve repair/replacement that require longer time spent on CPB and with the aorta cross-clamped [3,13]. Additionally, difficulty separating from CPB or the need to go back to full CPB support are also associated with increased risk of AKI. Anesthetic management and patient physiology that lead to hypoperfusion, hypovolemia or anemia, venous congestion, and need for inotropes further exacerbate the risk [20].

Administering intravenous colloid or crystalloid can optimize the circulation by increasing preload, which may increase stroke volume and cardiac output. Modern indicators of whether a patient’s circulation will benefit from additional fluid (so-called “fluid responsiveness”) are generally based on dynamic tests where cardiac output, stroke volume, or pulse pressure are measured during different loading conditions. Some common tests include passive leg raise where the venous blood that typically pools in a patient’s legs is returned to the circulation by leg elevation, and the resulting change in cardiac output is measured by echocardiography or esophageal doppler and pulse pressure variation (PPV) where larger changes in pulse pressure (and stroke volume) over the positive pressure respiratory cycle indicate that administering more fluid will increase cardiac output [21,22]. These goal-directed endpoints of fluid administration perform better than static indicators, such as central venous pressure or pulmonary artery occlusion pressure, at predicting improvement in cardiac output after fluid administration [23]. However, the data on perioperative fluid administration are mixed on whether a liberal or restrictive perioperative strategy is optimal; it likely depends on specific patient pathophysiology with conditions such as aortic stenosis, requiring higher cardiac filling pressures and more fluid resuscitation [24]. Fluid overload, often defined as gaining 10% above admission weight (e.g., an 80 kg patient gaining 8 kg or more of retained fluid weight), has long been known to be associated with adverse outcomes in the setting of AKI [25]. This holds true in CSA-AKI as well, where several studies have shown that there is an association between the degree of positive fluid balance (intra- and postoperative) and the development of postoperative CSA-AKI as well as an association with other adverse patient outcomes [26,27].

An emerging area in risk stratification is the combination of patient demographic factors and novel serum and urine biomarkers, rather than creatinine and urine output [28,29]. The success of many preventative treatment strategies rests on the ability to correctly identify patients at highest risk of developing AKI, and assess this risk prior to major changes in serum creatinine or urine output. Before delving into novel blood and urine biomarkers, there is emerging literature around the concept of renal reserve and its ability to predict the future development of CSA-AKI. At its core is the concept that at baseline, the kidney is not working at its maximum glomerular filtration rate (GFR), and that if stressed (e.g., with a large protein load or in the setting of pregnancy), the kidney will begin to hyperfilter, increasing the GFR by more than 20% [30,31]. In a series of recent papers, Ronco and colleagues have demonstrated that patients with an increased renal reserve (larger increase in GFR following a protein load) are less likely to develop postoperative CSA-AKI. They measured preoperative renal reserve with a protocolized protein load of 1.2 g/kg and demonstrated that in a cohort of 110 patients undergoing cardiac surgery, reserves were significantly lower in those who went on to develop CSA-AKI, and that preoperative reserves provided an area under the ROC of 0.83 (95% CI 0.70–0.96) for the future development of postoperative AKI [32]. While the physiology behind renal reserve is well grounded, future studies should investigate its impact on AKI risk and determine if certain interventions can reduce the incidence of AKI in those with diminished reserves.

### 1.2. Diagnosis and Biomarkers

The current definition of AKI is based on the Kidney Disease Improving Global Outcomes (KDIGO) criteria (Table 1). As previously mentioned, there were consensus definitions prior to KDIGO with the Risk, Injury, Failure, Loss, End-stage (RIFLE) and AKI-Network (AKIN) being the most commonly used and reported, both in the setting of CSA-AKI and other forms of kidney injury [33,34]. The KDIGO criteria meld these other two and allow for earlier disease detection by including smaller increases in baseline creatinine [35]. Though representing an advance toward standardizing AKI diagnosis, this definition still relies exclusively on evidence of functional changes of the kidney in serum creatinine and urine output. Thus, the current system fails to quantify the cellular and tubular injury that precedes drops in glomerular filtration, which results in low urine output and renal dysfunction. Newer biomarkers involved in tubular ischemia and cell cycle arrest that allow prediction of progression to AKI include neutrophil gelatinase-associated lipocalin (NGAL), insulin-like growth factor-binding protein 7 (IGFBP7), and tissue inhibitor of metalloproteinases-2 (TIMP-2). Taken together, IGFBP7 and TIMP-2 concentrations in urine have demonstrated predictive capability in cardiac surgical AKI and are now FDA-approved under the name Nephrocheck^®^ [8]. NGAL has been shown to independently predict duration of ICU and hospital length of stay, even when AKI was negative by creatinine measurements [6]. As these biomarkers become more widely used in clinical practice, they are likely to be incorporated into diagnostic criteria of AKI in the future [10]. An in-depth discussion of the benefits and limitations of these biomarkers of AKI are beyond the scope of this review and are discussed elsewhere within this edition.

### 1.3. Pathophysiology

The discovery and validation of new biomarkers are contingent on a more robust understanding of the pathophysiology of acute kidney injury, which is complex and heterogenous. While an in-depth discussion of the pathophysiology is beyond the scope of this paper, we will briefly discuss it here. At the organismal level, changes in hemodynamics play an important role in the development of CSA-AKI. Arterial hypotension, low perfusion from low cardiac output, elevated venous pressures, hypovolemia, anemia and hemolysis, ischemia and reperfusion, and changes from a pulsatile to non-pulsatile circulation during CPB all likely play a role [5] (Figure 1).

At the organ level, renal perfusion is highly controlled, since blood brings oxygen and nutrients but can also increase the filtration demands on the kidney. Higher blood flow requires more energy expenditure from ion transport pumps to main electrolyte gradients, increasing metabolic demand. The cortico-medullary junction, in particular, may be at elevated risk of ischemia given its low resting PO2 of 10–20 mmHg [36]. Any imbalance in oxygen delivery and consumption due to microemboli (atheroembolic or thromboembolic), arterial hypoxia, or low flow (due to arterial hypotension, venous hypertension, or neurohormonal vasoconstriction) can cause organ damage [37,38,39].

At the cellular level, the predominant mechanisms of injury include the interrelated cycle of inflammation and reactive oxide species (ROS) production. Contact of the circulating blood volume with the CPB circuit causes immune activation as evidenced by measurable increases in IL-6 (an inflammatory cytokine) and IL-10 (an anti-inflammatory cytokine), and, the higher the level of IL-6, the higher the corresponding risk of subsequent AKI [40]. In some studies, avoidance of CPB by “off-pump” techniques in CABG has been associated with lower rates of AKI, which will be discussed in more depth below [41,42,43].

Hemolysis, as a response to the extra-corporeal membrane and CPB circuit, also plays a role in CSA-AKI. It releases free hemoglobin which binds nitric oxide leading to vasoconstriction, increase ROS production, and induces heme oxygenase 1 expression which is associated with increased rates of AKI in experimental models and following cardiac surgery [44,45]. In a case-control study of 10 patients undergoing cardiac surgery who developed CSA-AKI compared to 10 risk-matched cardiac surgical patients who did not, free hemoglobin levels were more than double in the CSA-AKI group. [46].

## 2. Clinical Care

There have been several investigations into preventing the development of CSA-AKI by identifying those at high risk, and delivering algorithmic, bundled care [11,29,47]. These protocols aim to achieve an adequate intra- and postoperative circulation by an algorithmic approach to volume administration, pressors, and inotropes to avoid hypotension, venous congestion, and hypoperfusion [11,29,47,48,49]. Specific singular agent interventions have largely failed to prevent AKI; this is in part because of what we have labeled as CSA-AKI is likely a conglomerate of several pathophysiologic processes that all end in a common pathway of tubular injury (Table 4).

Once the diagnosis of CSA-AKI is established, specific treatments are lacking and limited to standard supportive critical care, adequate nutrition, glycemic control, and RRT [3]. However as discussed below, recent interventions that have combined early identification with biomarkers with kidney-focused care bundles have demonstrated promising outcomes [28,29].

### 2.1. Prevention

True primary prevention of AKI would be the most effective way to limit the subsequent harms associated with renal failure. However, while many different interventions have been studied, there is no definitive prevention for CSA-AKI [29,49] (Table 4). In general, despite physiologic plausibility and early success in animal models, all specific singular interventions have not reliably resulted in improvement in renal outcomes. These include pharmacologic agents, such as fenoldopam, so called “renal-dose” dopamine, sodium bicarbonate, acetaminophen, dexmedetomidine, propofol, N-acetylcysteine, and steroids; non-pharmacologic interventions, such as remote ischemic preconditioning; and changes in surgical technique, such as avoiding CPB or percutaneous valve replacement (Table 4). Flaws with these previous studies may have been the lack of standardized diagnostic criteria of AKI or the inability to accurately identify patients at high risk of AKI to appropriately tailor therapies.

To that end, Meersch, Zarbock et al. performed the PrevAKI trial, a single-center randomized controlled trial which identified cardiac surgical patients at high risk of AKI by using urinary TIMP-2 and IGFBP7 (Nephrocheck^®^) four hours after CPB (Table 5). In total, 276 patients were randomized to a control group where standard care included keeping MAP > 65 mmHg and CVP between 8–10 mmHG, and starting angiotensin converting enzyme inhibitors (ACEi) or angiotensin II receptor blockers (ARBs) once hemodynamics stabilized and the patient became hypertensive. The intervention group received a bundled care package consisting of strict KDIGO guidelines for avoidance of nephrotoxic agents, holding ACEi or ARBs for 48 h postoperatively, close monitoring of serum Cr and urine output, avoidance of hyperglycemia, consideration of alternatives to radiocontrast agents, and PiCCO catheter-guided algorithmic approach to hemodynamic and volume monitoring and management. The algorithm involved first administering crystalloid to a stroke volume variation of < 11, next adding dobutamine or epinephrine for a cardiac index of <3 L/min/m^2^, and finally using norepinephrine to achieve an MAP > 65 mmHg. Adherence to this protocol by the intervention group resulted in significantly more dobutamine (31.1% vs. 9.4%, *p* < 0.001), less hyperglycemia (50.7% vs. 75.4%, *p* < 0.001), and fewer ACEi/ARBs (10.9% vs. 30.4%, *p* < 0.001). The primary outcome was the occurrence of any stage KDIGO AKI at 72 h after surgery with secondary outcomes including AKI severity; mortality; RRT; persistent renal dysfunction (PRD; defined as a ≥0.5 rise in serum Cr from preoperative baseline); length of ICU stay and hospitalization; and a composite of death, RRT, and PRD. There was a statistically significant decrease in all-stage AKI in the intervention group compared to control (55.1% vs. 71.7%, *p* = 0.004, absolute risk reduction 16.6% with 95% CI 5.5–27.9%). KDIGO stage 2 and 3 AKI were also lower in the treatment group (29.7% vs. 44.9%, *p* = 0.009). There were no significant differences for any of the other secondary outcomes, which the authors hypothesized could be due to the relatively short follow-up of 90 d, the majority of AKI being mild or moderate (despite multiple observational studies correlating mild to moderate AKI with long term kidney decline), and most AKI being diagnosed based on oliguria (which has not been associated with poor outcomes in a specifically cardiac surgical cohort) [49]. This single-center study was used as a template for a multi-center study in PrevAKI 2, where the primary outcome was adherence to the bundled measures. Adherence was higher in the intervention group and led to increased use of crystalloid volume administration and dobutamine for a higher MAP than the control group. In the secondary outcomes, rates of moderate to severe AKI were lower in the intervention arm, but there were no statistically significant reductions in RRT or mortality. However, again, the study was underpowered with only 280 patients (Table 5) [29].

Another study identifying the importance of applying care bundles to patients at high risk of CSA-AKI was conducted by Engelman et al. They analyzed 435 patients before and 412 after the introduction of NephroCheck use to screen cardiac surgical patients (excluding those with preoperative Cr of >2.0 and on dialysis) on postoperative day one for risk of AKI. Different NephroCheck values were used for stratification. Values of <0.3 (ng/mL)^2^/1000 prompted normal care, “fast tracking” patients to stepdown units if they met other ICU discharge criteria. An intermediate group with NephroCheck values of 0.3–2.0 required hourly UOP monitoring, the avoidance of nephrotoxins (non-steroidal anti-inflammatory drugs, ARBs/ACEi, vancomycin, and gentamicin), repeat NephroCheck at 24 h, and, if the patient had urine output of <0.5 mL/kr/h for 3 h, the activation of an “acute kidney response team” (AKRT). NephroCheck values of >2.0 resulted in activating the AKRT, composed of intensivists, nephrologists, cardiac surgeons, nurses, and advanced practitioners. The AKRT used an algorithm based on the same KDIGO recommendations as the PrevAKI trial to avoid nephrotoxins, discontinue ACEi/ARBs, avoid hyperglycemia, closely monitor urine output, and manage hemodynamics. This was completed using an algorithmic approach to administer fluids, inotropes, and pressors to maintain a cardiac index > 2.5 L/min/M^2^, systolic blood pressure > 130 mmHg, mixed venous O2 > 60%, minimize serum lactate, and optimize echocardiographic parameters. The trial did not include information on the specific algorithm or differences in dosages of inotropes, pressors, and fluids between the two groups. The primary endpoint was the development of KDIGO stage 2 and 3 AKI by serum Cr only, which occurred in 10 patients (2.3%) before NephroCheck versus 1 patient (0.24%) after NephroCheck (*p* = 0.01). After propensity score matching, of 338 patients before and after the intervention, 8 (2.4%) had stage 2 or 3 AKI versus only 1 (0.3%; *p* = 0.04%). Importantly, given the low severe AKI rates in the cohort, they looked at secondary outcomes, such as total length of stay, cost, 30-day mortality, and 30-day readmission rate finding no significant difference. Without randomization or detailed data on differences in doses of fluid, inotrope, and pressor interventions, the authors concluded that this protocol is hypothesis generating, and needs to be tested a priori in a larger, multi-center randomized controlled trial (Table 5) [28].

Another area of interest in prevention of AKI is remote ischemic preconditioning (RIPC). Initially described for myocardial protection in the 1980s by Murry et al., the underlying mechanism involves using brief periods of ischemia to activate stress pathways, which attenuate injury when cells are subjected to subsequent ischemia [72]. Observations that ischemia acted on more distant tissues led to the concept of RIPC which has been demonstrated for other organs including the kidneys [73,74,75,76]. Multiple small clinical trials from 2007–2015 in various surgical populations suggested a benefit to renal outcomes from RIPC [77,78]. This led to several large, randomized, placebo-controlled trials published in 2015; the largest of these two, from Meybohm et al. and Hausenloy et al., comprised over 3000 patients undergoing cardiac surgery (CABG and/or valve repair and replacement) and failed to demonstrate any difference in outcomes, however the populations had relatively low rates of CKD and were not identified to be at elevated risk of AKI [67,68]. In an editorial in the *British Journal of Anesthesia* in 2020, Zarbock, Kellum et al. argued that RIPC may still be an effective preventative measure for perioperative AKI with better understanding of the interaction between the autonomic nervous system and RIPC. In the perioperative period, an anesthetized patient’s autonomic nervous system does not respond normally to any stimulus. Additionally, the autonomic nervous system of patients with cardiac dysfunction may alter the response to RIPC. Zarbock argues that RIPC may still be effective if it can be more appropriately tailored to individual patients and their autonomic nervous systems, perhaps warranting future investigation [79]. In a meta-analysis of RCTs, a sub-group analysis of RIPC in propofol-free anesthesia demonstrated a reduction in AKI (32.7% in RIPC vs. 47.5% in sham, RR 0.700, *p* = 0.014, 95% CI 0.527–0.930), though this difference did not persist in a Propofol-based anesthetic (23.7% in RIPC vs. 24.9%, RR 0.928, *p* = 0.39, 95% CI 0.781–1.102) [80]. This finding may be another suggestion of the importance of the interaction between RIPC and the autonomic effects of anesthetic agents.

### 2.2. Treatment

Treatment once a diagnosis of AKI is established is, at present, limited to standard supportive care, including, when appropriate, RRT. Though some experimental therapies, such as stem cell infusion and alkaline phosphatase have shown promise in animal models and septic humans, at present there have not been studies in the context of cardiac surgery [81,82,83,84]. Holding ACE-inhibitors and ARBs perioperatively is usually recommended due to associations with hypotension on induction of general anesthesia, but links to more significant outcomes, such as MI or AKI, are controversial [5]. In a recent meta-analysis of nine trials (five RCTs, four cohort studies) including 6022 patients on chronic ACEi/ARBs there was no difference in mortality (OR 0.97, 95% CI 0.62–1.51) or major adverse cardiac events (OR 1.12, 95% CI 0.82–1.52), though withholding ACEi/ARBs resulted in less intraoperative hypotension (OR 0.63, 95% CI 0.47–0.85) [85]. Avoiding nephrotoxins, such as NSAIDs is recommended in most AKI settings [1]. Though controversy remains about the impact of modern radiocontrast agents on renal function, using the lowest acceptable dose in a euvolemic patient for an indicated study or intervention is an acceptable approach [86,87]. Often the benefits, such as an interventional radiology procedure in an unstable, bleeding patient, will outweigh the risks of contrast exposure. In the case of bleeding, a timely intervention to minimize transfusion is especially important, since transfusion is associated with AKI [88,89,90].

Nutrition support is another aspect of modern critical care where AKI complicates clinical care. In the setting of AKI, especially if requiring RRT, clinicians must alter their traditional total intake goals as well as change their protein intake targets. Goals of at least 20–30 kcal/kg/day (with higher total calorie and protein requirements for patients on RRT) should be targeted by using enteral or parenteral support where appropriate [1,91]. Coupled with nutritional support is glycemic control, with the goal of a blood glucose concentration between 80 and 150 mg/dL [1].

In the case of severe AKI with oliguria, hyperkalemia, acidosis, or uremia, RRT is the therapeutic mainstay. The options of frequency, modality (either continuous or intermittent), and dose are not associated with outcome differences [3]. Continuous RRT provides more hemodynamic stability than intermittent treatments, and is often used in the critically ill population, but has never been consistently shown to improve patient outcomes. RRT in mechanical circulatory support has its own attendant complexities but most often a continuous technique is used [92]. Trials of early versus late timing of initiating RRT in AKI suggest a benefit to early initiation, as long as the patient population has a high likelihood of AKI progression [93,94,95]. However, a recent examination of RRT timing did not demonstrate improved patient outcomes in a large scale, international randomized controlled trial; in fact, early initiation (in a non-CSA-AKI specific population) was linked to a greater dependence on RRT 90 days later [96].

## 3. Outcomes and Cost

### 3.1. Outcomes

CSA-AKI is associated with significantly increased risk of morbidity and mortality in both the acute postoperative period and in the ensuing years following surgery [12]. The short-term morbidity associated with AKI includes infections, prolonged mechanical ventilation, strokes, and myocardial infarctions, all of which are associated with longer ICU and hospital length of stay [97,98]. In the near-term, post-discharge rehabilitation or healthcare requirements are more complex in patients with AKI, with 73.7% discharged to continuing care vs. 52.3% without AKI (*p* < 0.001) [13]. In addition to these other conditions, long-term sequelae of CSA-AKI include the development of incident chronic kidney disease (CKD) and progression of previously established CKD, including its most severe form, end-stage renal disease (ESRD) [99,100,101]. A 2012 systematic review and meta-analysis by Coca et al. attempted to quantify the risk of progression from AKI to CKD or ESRD. It included 13 studies involving patients from 1975–2005 in a mixed medical and surgical population (only one study specific to cardiac surgery), totaling nearly 1.5 million patients. The majority of the included studies were retrospective database analyses and had varying definitions of initial AKI, though follow-up was mostly over two years. They calculated pooled hazard ratios (HRs) of 8.8 (95% CI 3.1–25.5, *p* < 0.0001, Z = 4.02) and 3.1 (95% CI 1.9–5.0, *p* < 0.00001, Z = 4.58) for progression to CKD and ESRD, respectively, among patients with any definition and degree of AKI. Additionally, Coca et al. observed that increasing severity of AKI corresponded with increasing HR of CKD (mild AKI with HR 2.0, moderate AKI with HR 3.2, severe AKI with HR 28) and ESRD (mild with HR 2.3, moderate with HR 5.0, severe with HR 8.0). The Translational Research Investigating Biomarker Endpoints in AKI (TRIBE-AKI) cohort consisted of 1251 who underwent cardiac surgery at two Canadian study sites, with monitoring of renal function including urinary and serum biomarkers. A sub-group of 613 of these patients were monitored for progression to CKD over a median follow-up of 5.6 years, and demonstrated a similar pattern with increasing severity of AKI by AKIN criteria. Without in-hospital postoperative AKI, 23.8% of patients progressed to CKD, but with stage 2 or 3 AKI 50% progressed to CKD [102]. The progression to CKD after CTS-AKI represents another area where biomarkers may add additional prognostic information. In the TRIBE-AKI cohort, epidermal growth factor (EGF) and monocyte chemoattractant protein-1 (MCP-1) were strongly associated with the risk of progression to CKD [103].

For non-renal morbidity, Hansen et al. found a composite of myocardial infarction, heart failure, or stroke at 5 years after cardiac surgery to be 24.9% for patients with postoperative AKI versus 12.1% for patients without AKI [98]. Regardless of the outcome studied, increasing severity of AKI corresponds with an increasing likelihood of morbidity. Mortality is also highly correlated with CSA-AKI, both in the long and short term [4,12]. In their meta-analysis of global cardiac surgical patients, Hu et al. calculated in-hospital mortality to be over seven times higher in patients who developed AKI than in those who did not (10.7% vs. 1.37%), and mortality at 1 to 5 years to be over 2.5 times higher (30.0% vs. 11.9%) [4]. Small changes in Cr and duration of AKI also appear to be important prognostic factors [19,104]. In one single-center study even a small increase in Cr (Δ0.1 to 0.5 mg/dL) was associated with a threefold increase in 30-day mortality [104]. In a prospective observational study, Brown et al. found increased mortality in both the short and long term with a longer duration of AKI diagnosed by AKIN criteria. When using propensity-matching, they found an HR for death at five years of 1.71 (95% CI 1.37–2.13) with AKI of 1–2 days, 2.08 (95% CI 1.32–3.30) with AKI of 3–6 days, and 4.78 (95% CI 3.08–7.44) with AKI duration of >6 days [19]. Even complete resolution of AKI is not associated with a return to baseline mortality; in one single-center study, patients who had complete normalization of serum Cr following CTS-AKI still exhibited increased mortality at two years with a relative risk (RR) of 1.8 (*p* = 0.001) [99].

### 3.2. Cost

Mirroring the morbidity and mortality associated with AKI, the related costs are also increased in both the short and long term. Short-term costs are incurred from increased ICU and hospital LOS with the additional supportive interventions this entails (mechanical ventilation, antibiotic/pressor/intrope administration, RRT, etc.) [12,13]. In one study from Germany on TAVR patients, post-procedural stage 3 AKI was even more expensive than requiring placement of a second valve [105]. In a study of the largest in-patient registry of hospitals in the United States, Alshaikh et al. found an increase in hospital costs in post-cardiac surgical patients of almost $26,000 associated with AKI without RRT, and over $69,000 for AKI with RRT, leading to a total annual cost for AKI in cardiac surgical patients approaching 1 billion dollars [13]. Medium-term costs can be incurred from medically complex discharges; Hobson et al. found discharge to home after cardiac surgery occurred in 90% of patients without postoperative AKI versus only 68% of patients with AKI [97]. Long term costs related to associated comorbidities, such as cardiovascular disease and CKD or ESRD become even more difficult to calculate but are likely in the billions of dollars [12,13,106].

## 4. Conclusions

CSA-AKI is common in cardiac surgery and reflects a complex biological combination of patient pathology, perioperative stress, and medical management. Current diagnostic criteria, though increasingly standardized, are predicated on loss of renal function. The addition of diagnostic biomarkers to clinical practice may be able to identify patients at risk of renal injury earlier in their course. The accurate and timely identification of a high-risk population may in turn allow for bundled interventions to prevent the development of CSA-AKI, but further validation of these interventions is necessary. Once the diagnosis of CSA-AKI is established, treatment is limited to standard supportive care. The cost of CSA-AKI is difficult to accurately estimate, given the diverse ways in which it impacts patient outcomes, from ICU length of stay to post-hospital rehabilitation to progression to CKD and ESRD. However, with the global rise in surgical volume, including cardiac surgery, these costs are undoubtedly large and growing.

## Figures and Tables

**Figure 1 jcm-10-05746-f001:**
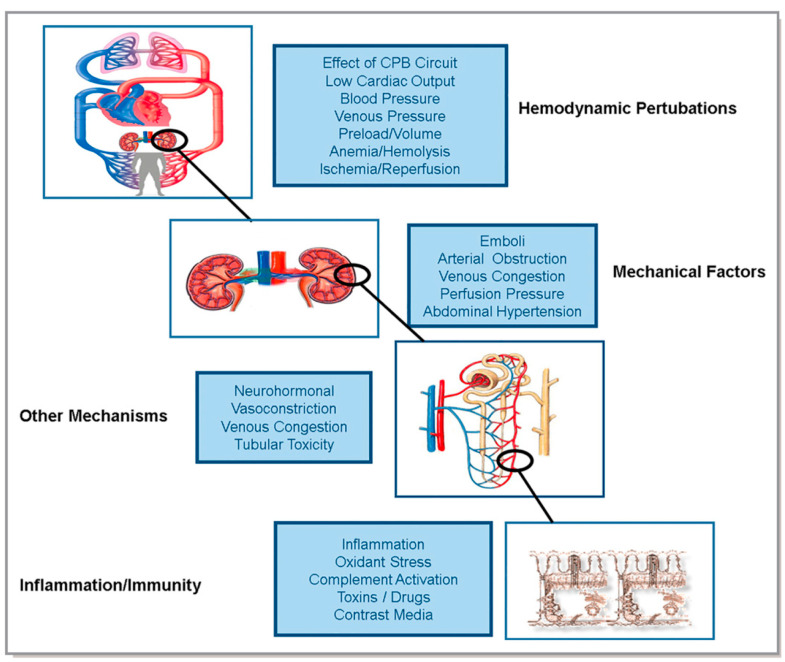
Pathophysiology of AKI (reproduced from ADQI by Nadim et al., 2018 [5]). From https://pittccmblob.blob.core.windows.net/adqi/20fig.pdf (accessed on 6 December 2021).

**Table 1 jcm-10-05746-t001:** KDIGO staging of AKI [1].

Stage	Creatinine	Urine Output
1	1.5–1.9 times baseline, OR ≥0.3 mg/dL	<0.5 mL/kg/h × 6–12 h
2	2–2.9 times baseline	<0.5 mL/kg/h for >12 h
3	>3 times baseline, OR >4 mg/dL, OR Initiation of RRT	<0.3 mL/kg/h for >24 h, OR Anuria > 12 h

**Table 2 jcm-10-05746-t002:** Risk factors for CSA-AKI [3].

Patient	Operative	Physiologic
Age	Surgical Complexity	Hypotension
Female Gender	CPB Duration	Inotrope exposure
HTN	Inability to separate from CPB	Hypovolemia
CKD	Low Hct during CPB	Venous congestion
Liver dz	Aortic cross-clamp time	Blood transfusion
PVD/CVA		Cardiogenic shock
Diabetes		Diuretic usage
Anemia		
Smoking		

**Table 3 jcm-10-05746-t003:** Scoring systems for AKI (adapted from [18]).

	Cleveland Score		Mehta Score		SRI Score	
	Definition	Points	Definition	Points	Definition	Points
Variable:						
Age	-	-	Varies	Varies	-	-
Race	-	-	Nonwhite	2	-	-
Sex	Female	1	-	-	-	-
Preop Renal function	SCr 1.2–2.1 mg/dL SCr > 2.1 mg/dL	2 5	Scr	Varies	GFR 31–60 mL/min GFR ≤ 30 mL/min	1 2
CHF	Yes	1	-	-	-	-
NYHA Class	-	-	IV	3	-	-
Diabetes	Insulin-requiring	1	Oral control Insulin-requiring	2 5	Any medication	1
COPD	Yes	1	Yes	3	-	-
MI ≤ 21 d ago	-	-	Yes	3	-	-
LVEF	<35%	1	-	-	≤40%	1
Previous Surgery	Yes	1	Yes	3	Yes	1
Preop IABP	Yes	2	-	-	Yes	1
Cardiogenic Shock	-	-	Yes	7	-	-
Surgical Timing	Emergency	2	-	-	Nonelective	1
Surgical Type	CABG only Valve only CABG + valve, other	0 1 2	CABG AV only AV + CABG MV only MV + CABG	0 2 5 4 7	Other than CABG or ASD only	1
Score Range		0–17		0–83		

SRI = simplified renal index, SCr = serum creatinine, GFR = glomerular filtration rate, CHF = congestive heart failure, NYHA = New York Heart Association, COPD = chronic obstructive pulmonary disease, MI = myocardial infarction, LVEF = left ventricle ejection fraction, IABP = intra-aortic balloon pump, CABG = coronary artery bypass grafting, AV = aortic valve, MV = mitral valve, ASD = atrial septal defect.

**Table 4 jcm-10-05746-t004:** Single-agent therapies for treatment of AKI.

	Therapy	Overview	Source	Outcome
Pharmacologic	Fenoldopam	Maybe less AKI but no change in overall outcomes	Meta-analysis * [50], N = 1107 pts in 7 RCTs	AKI (8.5% vs. 20.3%, RR 0.42 CI 0.26–0.69, *p* = 0.0006) but more hypotension (25.8% vs. 14.7%, RR 1.76 CI 1.29–2.39, *p* = 0.0003); no changes in mortality or RRT.
	Levosimendan	No impact	RCT, N = 849 [51]; RCT, N = 508 [52]	No difference in mortality or renal outcomes in either prophylactic administration or to treat low LVEF postop
	Dopamine	No impact	ICU Meta-analysis, N = 1019 in 24 studies (17 RCTs) [53]; RCT, N = 126 [54]	No impact on improvement of AKI in ICU patients; no change in incidence of AKI in cardiac surgical patients
	Spironolactone	No impact	RCT, N = 233 [55]	No difference in KDIGO AKI, trend toward harm (43% AKI in spironolactone group vs. 29% in placebo, *p* = 0.02)
	Bone morphogenetic protein-7 agonist (THR-184)	No difference	RCT, N = 452 [56]	KDIGO CTS-AKI rates similar in pts with recognized risk factors for AKI (range 74%–79% for various doses of THR-184 vs. 78% in placebo, *p* = 0.43)
	Mesenchymal stem cells (MSC)	No difference	RCT, N = 156 [57]	Pts with CSA-AKI got MSC v placebo with no difference in time to their Cr returning to preoperative baseline; median time to recovery was 15 d with MSC v 12 d with placebo (HR 0.81, CI 0.53–1.24, *p* = 0.32)
	Teprasiran (small interfering RNA)	Less AKI	RCT, N = 360 [58]	Pts at moderate to high risk of AKI by risk factors; all level AKI with teprasiran 36.9% vs. 49.7% in placebo (OR 0.58, 95% CI 0.37–0.92, *p* = 0.02)
	Sodium bicarbonate	No difference	Meta-analysis, N = 1092 in 5 RCTs [59]	Perioperative administration of sodium bicarbonate led to CSA-AKI rates of 42.6% vs. 41.3% in control (RR 0.95, 95% CI 0.74–1.22)
	Dexmedetomidine	May have AKI benefit, no mortality change	Meta-analysis, N = 1575 in 10 RCTs [60]	Perioperative administration of dexmedetomidine resulted in lower CSA-AKI rates (8.7% vs. 12.3%, OR 0.65, CI 0.45–0.92, *p* = 0.02) but similar mortality (0.8% vs. 2.3%, OR 0.43, 95% CI 0.14–1.28)
	N-acetylcysteine (NAC)	No benefit	Meta-analysis, N = 1391 in 10 studies [61]	Perioperative administration of NAC resulted in similar CSA-AKI rates (RR 0.841, 95% CI 0.691–1.023, *p* = 0.083)
	Statins	No benefit	RCT, N = 615 [62]	No change in CSA-AKI patients either naïve to or on preoperative statins; trial stopped early for futility; 20.8% AKI in statin group, 19.5% in placebo group (RR 1.06, CI 0.78–1.46, *p* = 0.75)
	Erythropoeitin	May benefit low-risk populations but overall no benefit	Meta-analysis, N = 473 in 6 RCTs [63]	Suggestion in sub-group analysis of reduced AKI when given prior to induction of anesthesia (OR 0.27, CI 0.13–0.54, *p* = 0.0002) and in low risk populations (OR 0.25, CI 0.11–0.56, *p* = 0.0008) but overall no difference (OR 0.69, CI 0.35–1.36, *p* = 0.28).
	Furosemide	No benefit	RCT, N = 126 [54]; RCT, N = 42 [64]	In Lassnigg, furosemide led to increase in Cr of 0.3 vs. 0.1 in the placebo group (*p* < 0.001). In Mahesh in high-risk patients it increased UOP (3.4 mL/kg/h vs. 1.2 mL/kg/h, *p* < 0.001) without changing AKI rates (43% vs. 43%, RR 1.1, 95% CI 0.6–2.2)
	Steroids	No benefit	RCT, N = 7286 [65]	High-risk patients undergoing CPB at given methylprednisolone (250 mg IV ×2) vs. placebo had CSA-AKI rates of 40.6% in steroids vs. 39.2% in placebo (ARR 1.04, 95% CI 0.96–1.11)
	Acetaminophen	May benefit	Retrospective cohort in pediatric cardiac surgery, N = 666 [66]	Postoperative acetaminophen exposure had a dose-dependent protective effect on CSA-AKI (OR for AKI 0.86 (95% CI 0.82–0.90) for each additional 10 mg/kg of acetaminophen
Technical	Remote ischemic Preconditioning	No benefit	RCT, N = 1612 [67]; RCT, N = 1385 [68]	See text
	Cardiopulmonary bypass avoidance (off-pump CABG)	No overall benefit	RCT, N = 2392 [43]; RCT, N = 2203 [69]	Garg demonstrated lower incidence of CSA-AKI in off-pump vs. on-pump CABG (17.5% vs. 20.8% for RR 0.83, 95% CI 0.72–0.97) but no difference in kidney function at 1 year (17.1% vs. 15.3%, RR 1.10 95% CI 0.95–1.29). Shroyer’s 5-year follow-up demonstrated decreased survival with off-pump CABG technique with a rate of death in the off-pump group of 15.2% vs. 11.9% in the on-pump group (RR 1.28, 95% CI 1.03–1.58).
	Percutaneous valve replacement	May benefit	Meta-analysis, N = 19,954 in 20 propensity-matched studies and 6 RCTs [70]; meta-analysis, N = 5536 in 6 RCTs [71]	Shah found lower AKI at 30 days after TAVR than SAVR (7.1% vs. 12.1%, OR 0.52, 95% CI 0.39–0.68) but similar incidence of RRT (2.8% vs. 4.1%, OR 0.78, 95% CI 0.49–1.25). Siddiqui included renal outcomes at 1 year and found no difference (OR 0.65, 95% CI 0.32–1.32).

* Unless otherwise marked, studies are in a cardiac surgical patient population. AKI = acute kidney injury, RCT = randomized controlled trial, RR = relative risk, CI = confidence interval, RRT = renal replacement therapy, LVEF = left ventricle ejection fraction, ICU = intensive care unit, KDIGO = kidney disease improving global outcomes, CSA-AKI = cardiac surgery associated-acute kidney injury, Cr = creatinine, HR = hazard ratio, OR = odds ratio, UOP = urine output, CPB = cardiopulmonary bypass, ARR = adjusted relative risk, CABG = coronary artery bypass grafting.

**Table 5 jcm-10-05746-t005:** Bundled care trials.

	Design	Outcome	Patients	Intervention	Results
PrevAKI 1 [49]	Single center prospective RCT	Primary: all KDIGO stage AKI within 72 h postop	276 patients (138 control and 138 intervention) undergoing on-pump cardiac surgery at high risk of AKI by Nephrocheck^®^ 4 h post-CPB	Bundled care including discontinuing ACEi/ARBs, avoiding nephrotoxins, and an algorithmic approach to hemodynamic management (see text) resulting in more dobutamine, less hyperglycemia, and fewer ACEi/ARBs in intervention group	Lower rate of all-stage AKI (71.7% in control vs. 55.1% in intervention, *p* = 0.004, OR 0.483, 95% CI 0.293–0.796)
PrevAKI 2 [29]	Multicenter prospective RCT	Primary: adherence to bundled care	278 patients (142 control and 136 intervention) undergoing on-pump cardiac surgery at high risk of AKI by Nephrocheck^®^ 4 h post-CPB	Bundled care including discontinuing ACEi/ARBs, avoiding nephrotoxins, and an algorithmic approach to hemodynamic management resulting in more dobutamine and more crystalloid in intervention group	Increased adherence to bundle (4.2% in control vs. 65.4% in intervention, *p* < 0.001, OR 42.92, 95% CI 17.61–104.60); secondary outcomes without difference in all-stage AKI (41.5% in control vs. 46.3%) but less stage 2 and 3 AKI (23.9% in control vs. 14.0%, OR 0.52, 95% CI 0.28–0.96)
Engelman [28]	QI initiative with pre- and post-implementation comparison	Primary: incidence of KDIGO stage 2 and 3 AKI	435 patients undergoing cardiac surgery before Nephrocheck^®^ use vs. 412 patients after	Activation of kidney response team in at-risk patients (based on Nephrocheck^®^) which advised targeted hemodynamic management, liberalized transfusion, and avoidance of nephrotoxins; no specific algorithms or in-group treatment differences reported	Lower stage 2 and 3 AKI after implementation (2.3% pre vs. 0.24% post, *p* = 0.01)

QI = quality improvement.

## Data Availability

Not applicable.

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
