# Peer review of "CSA-AKI: Incidence, Epidemiology, Clinical Outcomes, and Economic Impact"

_jcm, 2021, doi:10.3390/jcm10245746_

Round 1

Reviewer 1 Report

Drs. Schurle and Koyner have presented a very comprehensive review of cardiac surgery-associated AKI, touching on various aspects like pathophysiology, epidemiology, newer research and biomarkers, and impact. The table detailing trials done in the past is also noteworthy.

Their manuscript is detailed and covers most topics well, but this is a long review with a lot of information. I would recommend modifying the part about "prevAKI" trials into a small table so it is easier to read and understand, but also gets highlighted as it shows promising results.

Overall, the manuscript flows well, has good references, and touches on all aspects of CSA-AKI.

Author Response

Added a "Table 5" with a quick overview of the 3 bundled care trials discussed in the text.

Reviewer 2 Report

This is a comprehensive review of cardiac surgery associated AKI. The authors fully address the issues from epidemiology, pathophysiology, risk factors, treatment and prevention. This review could provide a holistic and evidence-based view to clinicians who participates or are interested in this area. There are only a few minor concerns.

1. Line 245-246: these two paragraphs are describing the prevAKI trial. Dividing it into two paragraphs may make the contextual continuity somehow confusing. The author could consider to merge the two paragraphs into one, with reduction of some details of the trial, for example, further condensing the part of exclusion criteria

2. Line 294 the paragraph for RIPC:

The two largest trials in 2015 NEJM mostly included patients without significant CKD:

  • Hausenloy et al: mean creatinine around 93-98 umol/L
  • Meybohm et al: % of CKD around 11% in both groups

Although the situation that required elective cardiac surgery could pose a high risk of AKI, the relatively low proportion of CKD in these studies still raise the concern whether the RIPC would be beneficial to patients with more advanced kidney disease?

Also, for RIPC:

The author may have interest in the following systematic review. May be the conlusion of this SR-MA could be placed into the table 4.

This SR-MA also reached similar conclusion after pooling studies with different sizes. Another interesting issue in this article is the subgroup analysis. The use or non-use of propofol as part of anesthesia seems to reach a statistically significant interaction on the AKI.  

Clinical Outcomes of Remote Ischemic Preconditioning Prior to Cardiac Surgery: A Meta‐Analysis of Randomized Controlled Trials. https://doi.org/10.1161/JAHA.116.004666

(3) Line 318 ....Avoiding nephrotoxins such as NSAIDs and ACE-inhibitors is recom-318 mended in most AKI settings [1]. ...

I suggest the authors avoid using the term "nephrotoxin" to cover ACE inhibitors. These may somehow misleading especially to the junior physicians and surgeons, who would genuinely take these agents (ACE-I or ARB) as a true nephrotoxin (its not....). This is also contrary to the nephro-protective effect of these agents in CKD care.

The debates between whether to stop or continue ACE-I or ARB before cardiac surgery remains controversial. The authors may have interest in briefly describe the controversy so far and provide some references to the reader.

Author Response

  1. Agree--consolidated the 2 paragraphs into 1. Took out some detail about exclusion criteria. Added a "Table 5" with a quick reference view of the 3 bundled care trials referenced in the text.
  2. Added caveat about low rate of CKD in 2 large RCTs of RIPC. Added meta-analysis and some discussion of the interesting sub-group analysis regarding propofol anesthesia.
  3. Split sentence into 2; 1 about the controversy of holding vs continuing ACEi/ARBs perioperatively and another about avoid nephrotoxins like NSAIDs.